# Dual Costimulatory and Coinhibitory Targeting with a Hybrid Fusion Protein as an Immunomodulatory Therapy in Lupus Nephritis Mice Models

**DOI:** 10.3390/ijms23158411

**Published:** 2022-07-29

**Authors:** Jordi Guiteras, Elena Crespo, Pere Fontova, Nuria Bolaños, Montse Gomà, Esther Castaño, Oriol Bestard, Josep M. Grinyó, Joan Torras

**Affiliations:** 1Experimental Nephrology Laboratory, Institut d’Investigació Biomèdica de Bellvitge (IDIBELL), L’Hospitalet de Llobregat, 08907 Barcelona, Spain; jguiteras@idibell.cat (J.G.); pfontova@idibell.cat (P.F.); 2Fundació Bosch i Gimpera, University of Barcelona, 08028 Barcelona, Spain; 3Experimental Nephrology and Renal Transplantation Laboratory, Nephrology Department, Vall d’Hebrón University Hospital, 08035 Barcelona, Spain; elena.crespo@vhir.org (E.C.); nuria.bolanos@vhir.org (N.B.); obestard@vhebron.net (O.B.); 4Pathology Department, Bellvitge University Hospital, Institut d’Investigació Biomèdica de Bellvitge (IDIBELL), L’Hospitalet de Llobregat, 08907 Barcelona, Spain; mgoma@bellvitgehospital.cat; 5Centres Científics i Tecnològics, L’Hospitalet de Llobregat, University of Barcelona, 08907 Barcelona, Spain; mcastano@ub.edu; 6Faculty of Medicine, Bellvitge Campus, L’Hospitalet de Llobregat, University of Barcelona, 08907 Barcelona, Spain

**Keywords:** costimulation, coinhibition, immunomodulation, CTLA4, PDL2, systemic lupus erythematosus, lupus nephritis, glomerulonephritis, autoimmunity, inflammation

## Abstract

Systemic lupus erythematosus is a complex autoimmune disorder mostly mediated by B-cells in which costimulatory signals are involved. This immune dysregulation can cause tissue damage and inflammation of the kidney, resulting in lupus nephritis and chronic renal failure. Given the previous experience reported with CTLA4-Ig as well as recent understanding of the PD-1 pathway in this setting, our group was encouraged to evaluate, in the NZBWF1 model, a human fusion recombinant protein (Hybri) with two domains: CTLA4, blocking the CD28—CD80 costimulatory pathway, and PD-L2, exacerbating the PD-1–PD-L2 coinhibitory pathway. After achieving good results in this model, we decided to validate the therapeutic effect of Hybri in the more severe MRL/lpr model of lupus nephritis. The intraperitoneal administration of Hybri prevented the progression of proteinuria and anti-dsDNA antibodies to levels like those of cyclophosphamide and reduced the histological score, infiltration of B-cells, T-cells, and macrophages and immune deposition in both lupus-prone models. Additionally, Hybri treatment produced changes in both inflammatory-related circulating cytokines and kidney gene expression. To summarize, both in vivo studies revealed that the Hybri effect on costimulatory-coinhibitory pathways may effectively mitigate lupus nephritis, with potential for use as a maintenance therapy.

## 1. Introduction

Systemic lupus erythematosus (SLE) is a highly complex and chronic autoimmune disease mostly mediated by B-cells, which release multiple circulating autoantibodies such as anti-double-stranded DNA (anti-dsDNA) [1,2]. The immune complexes formed can be deposited in multiple tissues, especially in kidneys, causing inflammation and tissue damage, leading to lupus nephritis (LN) and potentially to chronic renal failure and end-stage kidney disease. 

There are several murine models used to understand the cellular and genetic components of LN. A classical murine lupus disease develops in NZBWF1 mice, with a phenotype that resembles human lupus disease [3,4]. This strain spontaneously presents lymphadenopathy, splenomegaly, elevated serum anti-dsDNA antibodies and immune complex-mediated glomerulonephritis, leading to kidney failure and animal death [3]. As in human beings, in NZBWF1 mice lupus disease is predominant in females. This lupus disease model has been widely used to study the therapeutic efficacy of several immunosuppressive agents [5,6].

The MRL/lpr mouse strain, with a lymphoproliferative mutation and a defect in apoptosis, suffers from more severe lupus disease with more systemic involvement and nephritis [7]. This strain displays accelerated mortality in both males and females, and has higher concentrations of autoantibodies and immune complexes, including anti-dsDNA [7,8]. Furthermore, MRL/lpr mice specifically develop a full panel of lupus autoantibodies and have additional lupus manifestations such as arthritis, cerebritis, skin rash, and vasculitis [9]. Due to the early onset and exacerbated severity of the disease, MRL/lpr mice are also widely used for the assessment of candidate therapies for lupus [10,11].

The pathogenesis of SLE is caused by the appearance of anti-nuclear autoantibodies produced by long-lived plasma cells, which contribute to inflammation that causes cell damage and an increase in antibody titers, resulting in an auto-amplification loop [12,13]. This vicious circle is a key factor that can lead to the dysregulated production of aberrant type I interferon (IFN), resulting in a dysregulation of several pathways [14,15]. This ends up producing an activation of plasmacytoid dendritic cells by an increase in the expression of costimulatory molecules, directly activating effector T-cells and finally B-cells through B-cell-activating factor (BAFF) and a proliferation-inducing ligand (APRIL) [16,17]. This signaling cascade may affect kidneys by the release of anti-dsDNA autoantibodies by B-cells, which causes the deposition of immune complexes in the renal parenchyma and the activation of the complement system, especially in the glomeruli [18,19].

Standard treatments for SLE were based on nonspecific immunosuppressive agents such as corticosteroids and cyclophosphamide (CYP), and more recently Mycophenolate Mofetil. However, these current treatments may cause several side effects and serious infectious complications [20,21]. In recent years, many drugs have been studied for their potential to address different steps in this pathogenic cascade. Many are biological agents directed against specific targets, or costimulatory pathways, aiming to efficiently immunosuppress without the side effects caused by conventional immunosuppression drugs. Nifrolumab is currently an approved agent targeting the receptor for IFNa, indicated for SLE. Belimumab is also an approved agent targeting BAFF and indicated for LN. Voclosporine can also arguably be considered an LN-directed therapy due to its effect on T-cells [22,23,24]. Rituximab (anti-CD20), Eculizumab (anti-complement 5), Atacicept (anti-BAFF and anti-APRIL), and Abatacept (anti-CD80) are some of the other drugs that have significant effects on LN [9]. In this regard, it has been observed that treatment with Abatacept (CTLA4-Ig) and a suboptimal dose of CYP significantly prolonged survival, although without evidence of reduced glomerular immune complex deposition. Our group showed limited protection using CTLA4-Ig in the nephritis NZBWF1 model [21]. Moreover, Abatacept also failed to prevent lupus flairs in human lupus disease [1,25]. In addition, it was reported that CTLA4-Ig also has an effect on humoral response, as blocking the interaction between APCs and T-cells reduces the subsequent maturation of B-cells into plasmablasts, which will release circulating autoantibodies [26,27]. Blocking other costimulatory signals necessary for T-cell activation also appears to prevent disease progression in lupus mice [2,28].

The PD-1 signaling pathway has recently been suggested as a crucial target to prevent the development of SLE [29,30]. Studies showed that the PD-1 pathway regulates peripheral tolerance and protects tissues from autoimmune attack [31]. In addition, PD-1 generally reduces the affinity of the interaction between APC and T-cells in a self-reactive context rather than in a foreign antigen setting [32]. On the other hand, circulating soluble PD-1 (sPD-1) has also been reported as a marker of SLE, as high levels of this molecule are closely related to the evolution of the disease, as also found in other autoimmune diseases such as rheumatoid arthritis [33,34,35,36]. 

Taken together, these studies demonstrate the importance of costimulatory pathways in the development of SLE and the need to block the activation of both T-cells and B-cells to prevent the onset of the disease. In this context, our group designed Hybri, a fully humanized recombinant protein construct that consists of an IgG1 Fc linked to a CTLA4 molecule bound to two PD-L2 molecules by flexible polypeptide linkers. This recombinant protein construct dually binds the CTLA4 domain to CD80, blocking the costimulatory CD28–CD80/86 pathway, and the PD-L2 domain to PD-1, activating this coinhibitory pathway. Surface Plasmon Resonance (SPR) showed selective binding of Hybri to its targets CD80 and PD-1 simultaneously, also demonstrating that neither domain affects the binding affinity of the other. The study also showed Hybri’s therapeutic efficacy in rodent models of renal warm ischemia/reperfusion and renal allotransplantation [37]. 

In the present study, we assessed the dual and opposing coinhibitory targeting conferred by Hybri, which might promote favorable immunomodulatory mechanisms and help prevent the development of lupus nephritis. To additionally assess the effectiveness of Hybri in the autoimmune setting, we used the classical NZBWF1 model as well as the more severe MRL/lpr model.

## 2. Results

### 2.1. Hybri Effect in NZBWF1 Mice

Following SPR validation of the binding affinity of Hybri to its postulated mouse ligands, as reported elsewhere [37], the first set of experiments was carried out with a well-known strain of mouse widely used in many SLE-related studies. This strain is characterized by the production of anti-ds-DNA from approximately 16–20 weeks of age, causing especially glomerular damage in females, very similar to human, and with a mortality of 50% at 9 months of age. Administration of 200 µL intraperitoneal phosphate-buffered saline and Hybri 20 mg/kg doses resembled the therapeutic schedule of CTLA4-Ig used in the clinics (on days 0, 4, 14, 28, 56, and 84), while intraperitoneal CYP 50 mg/kg was administered every ten days. In the NZBWF1 model, overt renal disease appeared earlier in the vehicle group; at week 28, two mice had proteinuria above 3 mg per day. In two more mice, disease onset ccured before week 32. Two more mice subsequently developed high proteinuria levels, meaning a total of 75% of the animals were affected in the vehicle group. In the Hybri group, proteinuria above 3 mg per day was observed in one mouse at week 32 and in two more at week 36, equivalent to 33% of the group. The CYP group showed similar development of the disease, with one mouse presenting severe proteinuria at week 32 and another at week 36, equivalent to 22% of the group.

Regarding cumulative survival, 25% of the mice in the vehicle group died between the 32nd and 34th week of the study, as presented in Figure 1. In contrast, 100% of the mice in the two treated groups were alive at the end of the experiment.

### 2.2. Proteinuria and Anti-dsDNA Antibodies in NZBWF1 Mice

Proteinuria levels increased progressively in the vehicle group during the study, up to 337 mg/kg per day at week 28 and reaching 342 mg/kg per day at the end of the study (Figure 1). In the group treated with cyclophosphamide, proteinuria levels were constant and did not even reach 50 mg/kg per day. A similar trend was also observed throughout the study in mice treated with Hybri, with a slight increase in proteinuria levels at the end, reaching 121 mg/kg per day. Significant differences were observed in the proteinuria levels of the vehicle group with respect to the treated groups at both week 28 and in the last week of the experiment.

Throughout the study, albuminuria displayed similar trends to proteinuria in the three groups studied, with the vehicle group showing a progressive increase from week 24 to heavy values of 256 mg/kg per day at week 28, and as high as 275 mg/kg per day in the last week of the experiment. Albuminuria levels in the cyclophosphamide group did not increase, never exceeding 20 mg/kg per day. Finally, the levels in the Hybri group remained consistently low, with a minimal rise in the last week, reaching 33 mg/kg per day.

Regarding the anti-dsDNA antibodies, an increase was seen in all groups from 24 weeks of age, although the increase was more pronounced in the vehicle group, reaching a peak of 610 KU/mL at week 28 and remaining at a similar level at the end of the study. The increase in anti-dsDNA in the groups treated with Hybri and cyclophosphamide was significantly lower with respect to the vehicle group. Notably, levels of anti-dsDNA in Hybri mice were the same as in CYP-treated mice.

### 2.3. Renal Histopathology, Immunohistochemistry, and Immunofluorescence in NZBWF1 Mice

The histopathological study also revealed significant differences at the structural and inflammatory level between the untreated group and the treated groups. Whereas the vehicle group showed typical severe LN lesions, the Hybri group showed a clear reduction in glomerular deposits, extracapillary proliferation, and interstitial infiltrate, along with the absence of tubular atrophy and interstitial fibrosis. The mice treated with cyclophosphamide displayed minimal mesangial expansion and endocapillary proliferation (Figure 2).

Regarding renal infiltration of CD3^+^ cells (T cells), the group treated with Hybri showed a significant reduction in the number of interstitial migrated cells, with more than three times fewer migrated cells in some mice with respect to the vehicle group. As expected, the CYP group presented the lowest infiltrate score. Analysis of CD45R^+^ infiltrate (B-cells) showed that treatment with Hybri reduced the infiltrate by up to three times, while treatment with CYP further reduced it. Analysis of the interstitial F4/80^+^ (macrophages) revealed the same tendency as the other cell subsets, where the infiltrate in the Hybri and CYP groups was significantly lower than in the vehicle group. In the case of the CYP group, macrophage infiltration was minimal. Likewise, immunofluorescence studies of the glomeruli immune deposition showed a significant reduction of both renal IgG and C3 in both treated groups.

See Figure 3 for representative histological images.

### 2.4. Analysis of Peripheral Blood Cell Subsets with Flow Cytometry in NZBWF1 Mice

Peripheral blood cytometric analysis displayed some population differences between the vehicle and the treated groups (Table 1). The percentage of CD4^+^ T-cells was significantly reduced and CD8^+^ T-cells significantly increased in the CYP group compared to the vehicle group. Additionally, there was a reduction in PD-1 expression in the CD4^+^ subset compared to the vehicle group, whereas in Hybri-treated mice there was an increase in PD-1 expression in the CD4^+^ subset.

Lower expression of PD-L1 in antigen-presenting cells was observed in DCs in mice treated with CYP, paralleling the low expression of PD-1 in effector CD4 cells, while there was no change in Hybri mice. In contrast, expression of the PD-L2 ligand in DC was slightly lower in mice treated with CYP, with a clear increase in mice treated with Hybri, again paralleling the profile in CD4 cells expressing PD-1.

The Hybri group showed an increase in the percentage of stimulated monocytes over the other two groups, which could be given to a homeostatic balancing effect. However, Ly6C expression, which is an indicator of bone marrow cell recruitment, was shown to be lower than in the other groups. On the other hand, no increase in the percentage of stimulated monocytes was observed in the CYP-treated group, but there was an increase in Ly6C expression in both total and stimulated monocytes. Activation of monocytes may suggest the recruitment of this cell subset to the affected area, in this case the kidney.

As shown in Figure 4, differences among groups were observed in the Th17 cell subset, with the vehicle group showing a clear increase in this cell type. In this case, the expression of IL-17 was numerically higher than in the Hybri and CYP groups when referenced to the percentage in the CD4^+^ subset, especially due to the high expression of IL-17 in two of the five analyzed untreated mice.

Finally, no substantial differences in peripheral Tregs expression were seen in the Hybri-treated group compared to the vehicle. However, the CD25^+^ subset in Hybri was numerically higher than in the CYP group, probably indicating that Hybri induced a substantial effect through this regulatory subset. The vehicle group had a larger CD25^+^ subset although it was numerically lower than the Hybri group, probably representing a counterbalance in response to Th17 cell overexpression.

### 2.5. Hybri Effect in MRL/lpr Mice

After achieving promising results in the NZBWF1 model, we decided to explore the therapeutic effect of Hybri following the same dose schedule and groups in a more intricate and severe SLE mouse model such as MRL/lpr. This model is presented as a more systemic and accelerated animal model, with higher anti-dsDNA titers and also including cerebritis with cognitive dysfunction, skin rush, and vasculitis starting at 8 weeks of age. The expected survival at six months is 50% of the litter. Our results showed that symptoms in this model were more severe, with some mice developing lymphadenopathies. In this strain, disease onset in the vehicle group occurred earlier, with three mice developing proteinuria exceeding 3 mg per day at week 11, and three more mice at week 13. In contrast, mice treated with Hybri did not develop the disease until week 19, when one mouse developed proteinuria. At week 20 and 22, another two mice exceeded the threshold of 3 mg per day. It should be noted that in this model, Hybri delayed the onset of the disease by a few more weeks than in the MZBWF1 model. Additionally, cumulative survival of 62.5% was observed in the vehicle group, which was lower than in the NZBWF1 model, as shown in Figure 5. One mouse in the Hybri group died, with a cumulative survival of 83.3%. Cyclophosphamide-treated mice did not suffer any mortality during the study.

### 2.6. Proteinuria and Anti-dsDNA Antibodies in MRL/lpr Mice

Figure 5 shows that the mean proteinuria levels in the vehicle group increased sharply at 13 and 15 weeks of age. During this period, there were the three deaths in this group, occurring in mice with the highest levels of proteinuria. From then on, vehicle mean proteinuria was lower until week 17 with a subsequent but less pronounced increase until the end of the study. There was no increase in mean proteinuria levels in the treated groups during the study.

Trends in albuminuria were like those of proteinuria. In the vehicle group, there was a significant increase in the early stages, a large decline in the surviving mice, and a subsequent increase after week 17. Treatment groups experienced few changes throughout the study. In the Hybri group, there was a slight increase in the last week of the study. 

Anti-dsDNA levels in MRL/lpr mice were more than 10-fold higher than in the untreated NZBWF1 mice, thus confirming the greater serological aggressiveness of the model. Despite the high levels of anti-dsDNA release, both treatments numerically reduced the levels of autoantibodies. Differences between the Hybri and cyclophosphamide treatments were marginal.

### 2.7. Renal Histopathology, Immunohistochemistry, and Immunofluorescence in MRL/lpr Mice

The histopathological evaluation (Figure 6) showed more severe semi quantification of renal damage in untreated mice than in NZBWF1 mice, again confirming the greater severity of the model. Hybri therapy greatly reduced glomerular deposition, tubular atrophy, and interstitial fibrosis, with an overall significant decrease in inflammatory score from 9.7 to 3.8. CYP treatment led to a lower histological score to 1.3 of inflammatory score, with an absence of glomerular deposits, extracapillary proliferation, tubular atrophy, and interstitial fibrosis.

CD3^+^ cell infiltration was also greatly decreased in the group treated with Hybri, with a reduction of more than 10 cells per field with respect to the vehicle group. The CYP group had a similar but slightly smaller infiltrate compared to that of the Hybri group, with a mean of less than one cell per field. In addition, CD45R^+^ infiltrate was drastically reduced following both Hybri and CYP-based treatment, with significant differences compared to infiltrate from untreated mice. Macrophage recruitment was also significantly affected by both treatments, with the Hybri-treated group showing a reduction of almost half compared to the vehicle group. The immunofluorescence studies also showed a significant reduction in both C3 and IgG deposits in the glomeruli. Concerning IgG deposition, the Hybri group showed an even greater reduction with respect to the vehicle group than the CYP group. 

See Figure 7 for representative histological images.

### 2.8. Circulating sPD-1 Determination by ELISA in MRL/lpr Mice

Levels of sPD-1 were quantified in all surviving animals at the end of the study, and a significant reduction was observed in both treated groups (Figure 8). Thus, sPD-1 expression was reduced by half in the CYP group, while mice treated with Hybri presented levels up to four times lower than the levels in the vehicle group.

### 2.9. Circulating Immunoinflammatory Profile in Both Mouse Models

Peripheral blood analysis with Luminex in the NZBWF1 model (Table 2) showed a significant reduction in the immunoinflammatory profile evaluated by measuring cytokines and inflammatory mediators in both active therapeutic groups. Hybri mice showed a significant reduction in CD80, IL-6, RANTES, MCP-1, and TSLP compared to untreated mice. In addition, IL-12 was clearly reduced. The CYP group showed a similar reduction in these cytokines, but also a reduction in IL-10, IL-2, and RANKL. Some other cytokines showed a non-significant reduction, as in the case of IP-10 in the Hybri group, which showed a 50% reduction in expression, and CD137, where the levels for the Hybri group were like those in the CYP group.

In the MRL/lpr model, the results showed a different spectrum of cytokine changes. There was a large reduction in LAG3 and PD-L2 as well as TIM3 and TSLP expression. There was also a non-significant reduction in NIG and CD137 expression. The CYP group showed a similar reduction in expression of these cytokines, but also a reduction in RANTES, IL-12 and NIG expression. As for the rest of the cytokines, no significant changes were observed. Levels of cytokines in the vehicle group in the two models were similar, but fewer differences were seen in the MRL/lpr model following treatment, probably because of the intrinsic severity of this model. 

### 2.10. Renal Gene Expression Analysis in Both Mouse Models

An LN- and inflammation-related array of 47 genes was configured and then analyzed for both models [37,38]. After discarding four genes for technical reasons, a total of 43 genes were included in the analysis (see Appendix A). The NZBWF1 results showed that the expression of 28 genes was downregulated by CYP, and a total of 26 genes were downregulated by Hybri.

The MRL/lpr results showed the downregulation of 20 genes following CYP treatment. In the Hybri group, 28 genes were downregulated. No upregulated genes were observed in either treatment.

Among the downregulated genes in Hybri mice, 20 of them were modulated in both the NZBWF1 and MRL/lpr models, as shown in Appendix A. Subsequent analysis was conducted using the Pathwax II database, showing that Hybri treatment affected 30 human disease-related pathways, in which primary immunodeficiency and the PD-1–PD-L1 pathway were among the most significant. Hybri treatment also affected 20 organismal system-related pathways, with complement cascade, T- and B-cell receptor signaling, Th1, Th2, and Th17 cell differentiation, and NK cell-mediated cytotoxicity being the most relevant. It also affected nine environmental information processing pathways, including NF-kappa B, cytokine–cytokine interaction, MAPK, TNF, and JAK-STAT. Finally, Hybri treatment modulated five cellular process-related pathways as well as one metabolism-related pathway.

Furthermore, it was observed that the most relevant genes in these interactions were C3, Ptpn6, CD3e, CD14, and CD86, which appear heavily involved in costimulatory pathways and in the protein–protein interaction network, as reported in the FunCoup database, which fits with the proposed mode of action of Hybri.

## 3. Discussion

This study of two different models of lupus nephritis demonstrates the clinical protection conferred by dual and opposing costimulatory targeting. In previous studies, our group showed that Hybri successfully inhibits the T-cell response and, although it is a human protein with partial homology to mice, it has good affinity for the postulated ligands [37]. After achieving good results with Hybri in the warm ischemia and allogeneic setting [37], we decided to assess its potential efficacy in the autoimmune setting. A clear clinical effect was observed in the NZBWF1 mouse model, a widely used and described LN model. Untreated mice showed a survival of 75%, as expected in the model, with no mortality in either treated group. Mice treated with Hybri showed levels of proteinuria like the group treated with cyclophosphamide, but with a slight increase at the last checkpoint. However, these final levels were still well below those of the untreated group. It may be argued that a minimum therapeutic regimen was used, which was based on the use of CTLA4-Ig in rheumatoid arthritis in clinics but, in this case, without the addition of prednisolone [39,40]. The formation of antibodies against the injected protein itself due to the lack of homology between species could be another alternative [41]. However, the selective mechanism of action of Hybri, targeting costimulatory molecules, confers a similar therapeutic effect without the classical toxic effects of cyclophosphamide. In this regard, future directions should include dose-response curves for Hybri, as well as test a more frequent dosage to determine whether responses to nephritis may be as effective as CYP.

The reduction in anti-dsDNA antibodies, as happens following CYP treatment, accompanied by the solid reduction in glomerular IgG and C3 deposition, suggests modulation of the B-cell response by Hybri. This effect on the humoral component has also been described in clinics following renal transplantation and treatment with Belatacept, a mutant second-generation Abatacept [27]. Indeed, the reduction in B-cells suggests the local reduction of potential anti-dsDNA-releasing cells. Additionally, the reduction in infiltrating macrophages and T-cells in the kidney suggests that the coinhibitory effect of the protein reduces cell interaction, preventing its activation and recruitment to the affected organ. It is also a good proof of principle to observe that mesangial expansion and endocapillar proliferation are not affected by Hybri treatment in the study. This would not be expected with a lymphocyte-specific targeted therapy. Hybri may also interfere with costimulation within the organ itself, as there are several cell types in the kidney that express some costimulatory molecules. There are, for example, renal parenchymal DCs expressing PD-L1 and PD-L2, podocytes expressing CD80 and PD-L2, or tubular cells expressing PD-L1 [42,43,44]. These findings indicate that Hybri exerts a global immune modulation effect, as it acts on the humoral and cellular arms.

Peripheral blood cell subpopulations underwent some changes, but two significant variations may explain the mechanisms through which Hybri improves the fate of the disease. The reduction in PD-L2 expression in DCs under CYP treatment in contrast with the increase caused by Hybri could be the consequence of different pharmacodynamic mechanisms in the two treatments. It is plausible that cyclophosphamide treatment provides an immunological environment conferred by the nonspecific reduction of several cell subsets, including the DCs PD-L2^+^ and CD4 PD-1^+^. In contrast, the Hybri target involves selective activation of the inhibitory PD-1–PD-L2 dyad, both in antigen-presenting and in effector cells. Findings in PD1-expressing cells or its ligands in mice under Hybri treatment in our study support this conclusion, with overexpression of PD-1 in the CD4^+^ subset and a clear increase in PD-L2 ligand in DC mice. As is the case in the vehicle group, a high expression of PD-L2 in the DCs of mice with lupus nephritis may be given to a homeostatic factor, where this pathway is possibly overregulated to counteract this development of the disease [45]. The group treated with Hybri showed an even higher expression in this cell type, suggesting that it enhances precisely this protective role that does not appear to be sufficient without treatment. This might mean that the drug reached its specific target, at least at the peripheral level.

An increase in Th17 cell numbers was seen in the untreated group. Although the effect of Th17 cells and even their close relationship to Tregs and Th1 cells is still not entirely clear [46,47,48], the changes in these cell subpopulations seen in our study support the hypothetical interrelation between these two populations. Thus, the higher expression of CD25 in CD4 T-cells seen in the vehicle and Hybri groups, indicative of the presence of circulating Tregs, may represent two different mechanisms. On the one hand, the increase in CD25 in untreated animals may correspond to a physiological counterbalance to Th17 overexpression in the disease, or to other effector cell mechanisms. This cell counterbalance aims to overcome the deleterious mechanisms of the disease, partly exerted by the Th17 clonal expansion. On the other hand, the increase in CD25 in the Hybri group might be related to the selective action of our dual protein on T-cell differentiation towards a regulatory arm. This is supported by the lack of Th17 cell expansion, indicating suppression of the effector arm of the immune response. Finally, CYP treatment clearly reduced the Th17 clonal expansion, but also reduced the number of regulatory CD25 cells. Again, this indicates that CYP has a non-selective mode of action, indiscriminately suppressing all cellular compartments, in contrast to the proposed selective mode of action of Hybri.

Of note, Hybri treatment produced substantial changes at the level of circulating cytokines. Most of the cytokines found at reduced levels following Hybri treatment are closely involved in the inflammatory context of the disease. Cytokines such as RANTES, MCP-1, and IP-10 promote the activation, maturation, and recruitment of proinflammatory cells in the area of inflammation, such as T-cells, DC, NK, and macrophages, among others. IL-6, CD137, and IP-10 also play an important role in the release of IL-2 and IFNγ and other cytokines specific to the disease cascade. The systemic shift from an intensive proinflammatory cytokine profile towards a smoother inflammatory setting supports one of the beneficial pharmacological effects of our compound. These results were like those in the CYP-treated animals, indicating that Hybri treatment modulates the inflammatory milieu in the same way as CYP, leading to disease attenuation.

A subsequent experiment including the same groups was performed using a different lupus-prone model. This model has a more accelerated mortality ratio [8], as reflected by our cumulative survival results compared to the previous model. This strain also displays a high concentration of circulating immunoglobulins as well as anti-dsDNA antibodies among other circulating autoimmune complexes. Our data confirm that the levels of anti-dsDNA are more than 10 times the levels of the NZBWF1 model [9,49]. The strong systemic alterations in this strain make the model more challenging when evaluating the phenotypic changes and protection offered by new treatments. Our results in this MRL/lpr model prove this more systemic and severe disease, in which the kidney is also more affected. Despite the differences in severity of these two murine lupus-prone models, both strains proportionally benefited from Hybri and CYP therapies, as demonstrated by the reduction in proteinuria, renal histological damage, and serological activity. 

However, the overall results showed that the inflammatory microenvironment in the MRL/lpr strain is more controversial and probably depends on other factors. It should be noted that this strain may also present high concentrations of rheumatoid factors, which cause an autoimmune disease where there is an over-release of sPD-1 [34,36]. As a close relationship between circulating sPD-1 levels and disease involvement has been described in this model, we further analyzed sPD-1 levels [33,35]. A clear reduction in sPD-1 levels was found, possibly because of the improvement in disease progression due to treatment, but it may be related to the linking of sPD-1 with Hybri PD-L2 domains, thus masking the circulating levels of sPD-1. Animals treated with CYP presented low levels of sPD-1, but Hybri-treated animals showed levels about half those in the CYP-treated group. The low levels of sPD-1 might be due to the reduction in the inflammatory milieu, as caused by CYP, and on the other hand by the ligation of the Hybri PD-L2 domain to circulating PD-1. In this intricate setting where an excess of sPD-1 could condition the Hybri effect, an increase in either the dose or the frequency of Hybri administration might result in an even better therapeutic effect. 

The information obtained from the genetic analysis showed that treatment with Hybri may also modify inflammatory and immunomodulatory pathways in the kidney. The genes analyzed in our study were previously evaluated in similar in-house models and other reported models closely related to the costimulatory pathways [36,37]. Most of the array genes related to a wide-ranging spectrum of pathways were modulated by cyclophosphamide, as previously reported [38]. Interestingly, Hybri treatment ameliorated renal disease through a group of genes like those modulated by CYP treatment, despite the different mechanisms of the two drugs. In this regard, Hybri modulated some human disease-related processes involving a myriad of immune mediators in which Th17 cell differentiation or NK cell-mediated cytotoxicity are present. Several affected genes also appear to be heavily involved in costimulatory pathways, which may be linked with the selective mechanism of action of our dual inhibitory protein.

To summarize, both in vivo studies exhibited that the Hybri protein may effectively mitigate lupus nephritis disease as an initial treatment and in maintenance therapy. It should also be noted that histopathology and changes in gene array analysis, as well as in the rest of the results obtained from procedures with samples from euthanasia in the vehicle control group could be underestimated, due to the fact that the mice with the worst nephritis were probably the ones who died before euthanasia and histopathological evaluation. In contrast with the poor results historically obtained with CTLA4-Ig and the hierarchical importance of the PD-1 coinhibitory pathway in this setting, it is possible that there is positive synergy thanks to the dual mode of action of Hybri. By inhibiting a costimulatory pathway and activating a coinhibitory signal, our hybrid fusion protein construct could promote several immunomodulatory mechanisms, helping to protect the kidney from the immunoinflammatory state of SLE.

## 4. Materials and Methods

### 4.1. Mice, Study Design, and Follow-Up; SLE Setting in NZBWF1 Mice and MRL/lpr Mice

Five-month-old NZB/WF1 female mice (The Jackson Laboratory, Charles River, Wilmington, MA) were randomly assigned into three groups: CYP: intraperitoneal Cyclophosphamide, 50 mg/kg (*n* = 8); Hybri: intraperitoneal Hybri protein, 20 mg/kg (*n* = 9); vehicle: intraperitoneal Phosphate-Buffered Saline, 200 µL (*n* = 8) as control. At six months of age, treatment was initiated. Hybri doses were defined by extrapolating the doses used clinically for rheumatoid arthritis. Being a protein with a molecular weight approximately twice that of Abatacept, twice the mass was defined as a dose.

Eight-week-old MRL/lpr female mice (The Jackson Laboratory, Bar Harbor, ME) were randomly assigned into three groups. CYP: intraperitoneal Cyclophosphamide, 50 mg/kg (*n* = 6); Hybri: intraperitoneal Hybri protein, 20 mg/kg (*n* = 6); and vehicle: intraperitoneal Phosphate-Buffered Saline, 200 µL (*n* = 8) as control. At ten weeks of age, treatment was initiated. 

Cyclophosphamide doses were administered every 10 days. Hybri protein and PBS doses were administered on days 0, 4, 14, 28, 56, and 84. Mice were treated for 12 weeks in both sets of experiments. Body weight was determined weekly from the beginning to the end of follow-up. Mice were placed in metabolic cages to collect 24 h urine specimens before the onset of treatment, and at regular times. Blood was obtained from the tail vein at monthly intervals. Kidneys were processed for histological and biochemical studies at the end of the study or at death. The studies ended at 36 weeks of age for the NZBWF1 strain and at 22 weeks of age for the MRL/lpr strain (Figure 9). 

The experiments were carried out in accordance with current EU legislation on animal experimentation and were approved by “CEEA: Animal Experimentation Ethics Committee”, the Institutional Ethics UB Committee for Animal Research, and the Animal Experimentation Commission of the Generalitat de Catalunya (Catalonian Government). Mice were housed in a room at constant temperature with a 12-h dark/12-h light cycle, were given free access to tap water, and were fed a standard laboratory diet. The criteria for animal euthanasia before the end of the study was based on decreased animal weight and overall appearance, suggesting worsening of the disease and symptoms of animal suffering.

### 4.2. Proteinuria, Albuminuria, and Renal Function

Twenty-four-hour urinary protein was determined by pyrogallol red reaction (Olympum Autoanalyzer AU400, Hamburg, Germany) in the Veterinary Clinical Biochemistry Laboratory of Universitat Autonoma de Barcelona. Twenty-four-hour urinary albumin was determined using a commercially available ELISA KIT (Active motif, Carlsbad, CA, USA) according to the manufacturer’s instructions. The intensity of the fluorescent signal is directly proportional to the albumin concentration in the sample.

### 4.3. Serum ELISA for Anti-DNA Antibodies

Levels of anti-DNA antibodies were measured using a commercially available ELISA kit (Alpha Diagnostic International, San Antonio, TX, USA) according to the manufacturer’s instructions.

### 4.4. Renal Lupus Histopathology

For histological analyses, 1–2 mm-thick coronal slices of kidney were fixed in 4% formaldehyde and embedded in paraffin. For light microscopy, 3–4 µm thick tissue sections were stained with hematoxylin and eosin, and periodic acid-Schiff. To determine the extent of renal damage, two blinded pathologists analyzed all kidney biopsies. Typical glomerular active lesions of lupus nephritis were evaluated: mesangial expansion, endocapillary proliferation, glomerular deposits, extracapillary proliferation, and interstitial infiltrates, as well as tubulo-interstitial chronic lesions, tubular atrophy, and interstitial fibrosis. Lesions were graded semi-quantitatively using a scoring system from 0 to 3 (0: no changes, 1: mild, 2: moderate, 3: severe). Finally, a total histological score was derived from the sum of all the described items.

### 4.5. Renal Immunohistochemistry Studies

Paraffin tissue sections were stained for CD3 (Abcam, Cambridge, UK), CD45R (Invitrogen, Waltham, MA, USA), and F4/80 (Hycult Biotech, Uden, The Netherlands). Sections were dewaxed with xylene and rehydrated with a decreasing battery of ethanol solutions until distilled water. Then, sections were blocked and immunoperoxidase labeled using a EnVision Dual Link Kit (Dako/Agilent, Santa Clara, CA, USA) in accordance with the manufacturer’s protocol. Peroxidase-conjugated secondary antibody staining was followed by diaminobenzidine substrate development, controlling the incubation time under the microscope. To quantify CD3 and CD45R expression, at least 15 high-power fields were counted, and the mean value was expressed. For quantifying F4/80 expression, a semiquantitative intensity score from 0 to 3 was used. 

### 4.6. Renal Immunofluorescence Studies

For the NZBWF1 model, slices of kidney were fixed in 4% paraformaldehyde, embedded in OCT Tissue Tek compound (Sakura, Alphen aan den Rijn, The Netherlands), and stored at −80 °C. Fluorescent staining of 5µm cryostat sections was used for confocal microscopy to quantify glomerular C3 and IgG deposition. Sections were directly stained with an FITC conjugated C3 (Nordic Immunology, Susteren, The Netherlands) and FITC conjugated goat anti-mouse IgG (Sigma-Aldrich, San Luis, MO, USA). For the MRL/lpr study, fluorescent staining of 5µm paraffin sections was used for confocal microscopy to quantify glomerular C3 and IgG deposition. Sections were stained with an FITC conjugated C3 (Nordic-MuBio, Susteren, The Netherlands) accompanied by a special amplification using the TSA Kit with Tyramide-AF568 (Invitrogen, Waltham, MA, USA). To perform the IgG stain, the FITC conjugated goat anti-mouse IgG (Nordic Immunology, Susteren, The Netherlands) was used. For analysis of C3 and IgG deposition, at least 10 glomeruli were visualized and photographed with an immunofluorescence confocal microscope (Leica TCS-SL spectral microscope, Mannheim, Germany). Fluorescence was quantified and normalized with Simulator-Leica confocal software and expressed as mean fluorescence intensity.

### 4.7. Phenotypic Peripheral Blood Population Analysis by Flow Cytometry

Blood samples were collected in heparin tubes and separated in cytometry tubes to add the antibodies and lysis buffer to analyze the different composition of immune cells. Cytometry tubes were incubated in the dark (25 min, RT) with antibodies. Study of populations was performed by using a BD FACS Canto II Cytometer and analyzed by BD FACSDiva ™ v9.0 Software (BD Biosciences, San Jose, CA, USA). BD FlowJo™ v10.8.1 Software (BD Biosciences, San Jose, CA, USA) was used for the generation of cumulative graphs.

To characterize the different cell populations, different antibodies were used. For the T cells tube: CD3 APC (Clone: 145-2c11) (BD 553066), CD4 FITC (Clone: RM4-5) (BD 553047), CD8a PE (Clone: 53-6.7) (BD 553033), PD-1 PerCP Cy5.5 (Clone: 29F.1A12) (Biolegend 135208); For the Dendritic cells tube: CD11c FITC (Clone: HL3) (BD 553801), PDL-1 APC (Clone: MIH5) (BD 564715), PDL-2 PE (Clone: TY25) (BD 557796); For the Monocytes tube: CD11b APC Cy7 (Clone: M1/70) (BD 557657), CD115 Vio Bright FITC (Clone: AFS98) (MILTENYI 130-105-162), Ly6C PE (Clone: 1G7.G10) (MILTENYI 130-102-391); For the Th17 tube: CD3 APC (Clone: 145-2c11) (BD 553066), CD4 FITC (Clone: RM4-5) (BD 553047), IL-17a APC Cy7 (Clone: TC11-18H10) (BD 560821); and for the Tregs tube: CD3e APC Cy7 (Clone: 145-2c11) (BD 557596), CD4 FITC (Clone: RM4-5) (BD 553047), CD25 PE (Clone: PC61) (BD 553866), FoxP3 APC (Clone: FJK-16s) (eBioscience 17-5773-82). All antibodies were provided by BD Biosciences, San Jose, CA; Biologend, San Diego, CA, USA; and Milteny Biotec, Bergisch Gladbach, Germany; conveniently titrated, mixed, and formulated for optimal staining performance. The permeabilization buffer (00-8333-56) and Fixation/Permeabilization Diluent (00-5223-56) (eBioscience, San Diego, CA, USA) were used in the Tregs tubes in order to permeabilize the cell nucleus and staining the FoxP3 transcription factor.

### 4.8. Serum ELISA for sPD-1

Levels of serum sPD-1 were measured using a commercially available ELISA kit (MyBioSource, San Diego, CA, USA) according to the manufacturer’s instructions. Results were calculated from the calibration curves and expressed in ng/L.

### 4.9. Measurement of Serum Levels of Immune and Inflammatory Factors by Luminex Fluorescent Assay

The quantification of CD137, CD80, IL-10, IL-12/IL-23p40, IL-17a, IL-2, IL-6, IP-10 (Cxcl10), LAG3, MCP-1, NIG, PD-L2, RANKL, RANTES, TIM3, TNF alpha, and TSLP concentrations in serum was conducted using Luminex ProcartaPlex ™ Multiplex Immunoassay (Thermo Fisher Scientific, Waltham, MA, USA) following the manufacturer’s instructions. Results were calculated from the calibration curves and expressed in pg/mL.

### 4.10. RNA Extraction, Reverse Transcription and Gene Expression Analysis: TaqMan Low Density Array Microfluidic Cards Quantification

For molecular studies, the kidney was immediately snap-frozen in liquid nitrogen and stored at −80 °C. RNA was extracted from the kidney with PureLinkTM RNA Mini Kit (Invitrogen, Waltham, MA, USA) according to the manufacturer’s instructions. RNA purity was analyzed on a NanoDrop (NanoDrop ND-1000V3.3, Wilmington, DE, USA). All samples had an A260/280 ratio > 2. RNA was stored at −80 °C. A total of 1000 ng of RNA was used to perform the reverse transcription using a High-Capacity cDNA Reverse Transcription Kit (Applied Biosystems, Warrington, UK) following the manufacturer’s instructions. 

Kidney tissue expression of immune-inflammatory mediators was quantified by TaqMan Low-Density Array (TLDA) microfluidic cards (Applied Biosystems, Warrington, UK). The TLDA were loaded, sealed, and run as prescribed in the protocol. qPCR was run in a 7900HT Fast Real-Time PCR system (Applied Biosystems, Warrington, UK). Cq values were calculated using the SDS software v.2.3 (Applied Biosystems, Warrington, UK) using automatic baseline settings and threshold. Then, 18 s was used as endogenous control. Controls, which were composed of distilled water, were negative for target and reference genes. Variations in gene expression was determined using the equation 2^−ΔΔCq^. A pathway analysis of single gene sets was set up using the online PathwaX.sbc.su.se web server, which applies the BinoX algorithm to KEGG pathways [50] and FunCoup networks [51].

### 4.11. Statistical Analysis

Overall cumulative survival was analyzed with the Kaplan–Meier method and cumulative overt disease were performed to compare groups with Log Rank Mantel–Cox test. One-way analysis of variance (ANOVA) with post hoc tests was performed to compare proteinuria, albuminuria, and anti-dsDNA antibodies throughout the follow-up, and gene expression and circulating cytokines at sacrifice. To compare histological data, the non-parametric Kruskal–Wallis test was used. *p*-value < 0.05 was considered significant. Data are expressed as mean ± S.E.M.

## Figures and Tables

**Figure 1 ijms-23-08411-f001:**
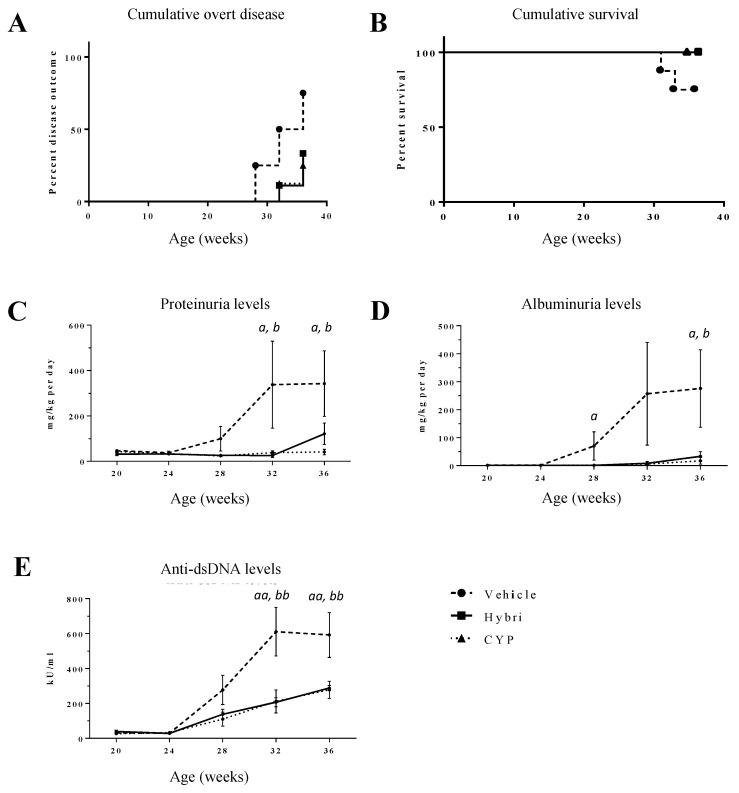
Hybri administration delayed the onset of overt disease, improved the survival curve, and preserved the functional renal parameters similar to those of mice treated with CYP throughout the study in the NZBWF1 mice model. (**A**) Cumulative overt disease for the NZBWF1 until the endpoint of the study at 36 weeks of mice age. Disease outcome was considered when proteinuria levels exceeded 3 mg per day. (**B**) Kaplan–Meier survival curve for the NZBWF1 model. Log Rank Mantel–Cox test revealed non-significant differences between groups neither for cumulative overt disease nor cumulative survival. Proteinuria (**C**) and albuminuria (**D**) values throughout the study are expressed in mg normalized per body weight per day. Serum anti-dsDNA (**E**) levels throughout the study are expressed in KU/mL. Vehicle group: *n* = 8; Hybri group: *n* = 9; CYP group: *n* = 8. Data are expressed as mean values ± S.E.M. Hybri vs. Vehicle: *a* = *p* < 0.05, *aa* = *p* < 0.01; CYP vs. Vehicle: *b* = *p* < 0.05, *bb* = *p* < 0.01.

**Figure 2 ijms-23-08411-f002:**
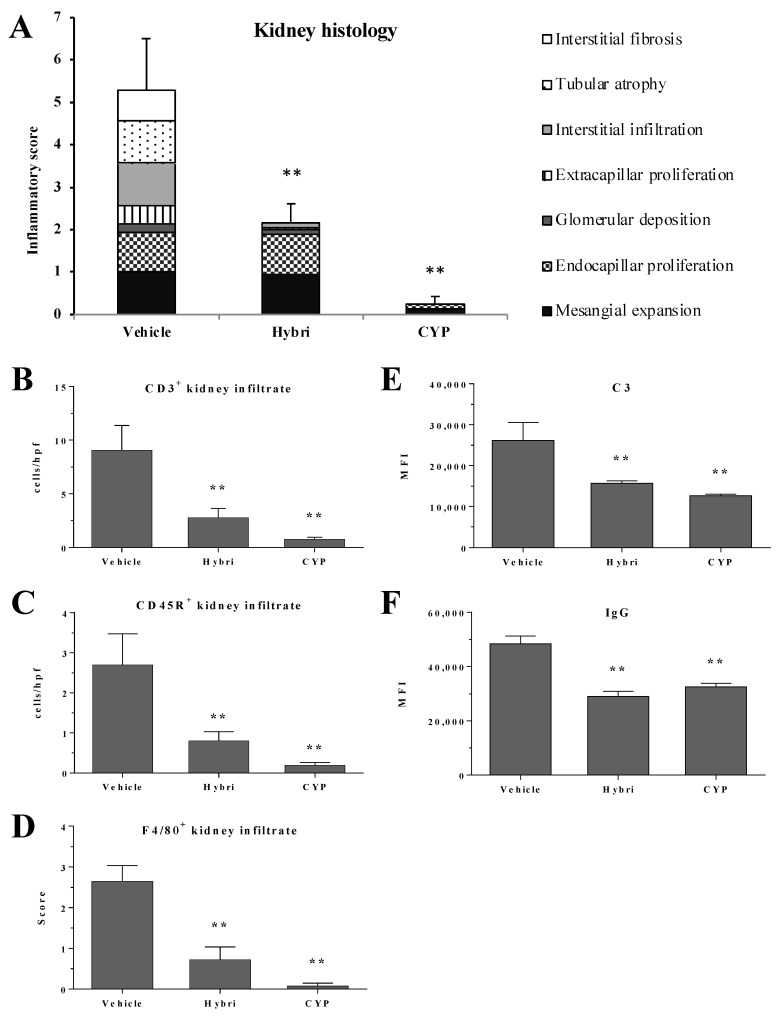
Hybri administration reduces morphological kidney damage and effector cell migration in the NZBWF1 mice model. Analysis of samples from surviving mice at the end of the study, at 36 weeks of age. (**A**) Renal histopathology parameters assessed by semiquantitative inflammatory score for hematoxylin/eosin and periodic acid-Schiff staining. Vehicle group: *n* = 6; Hybri group: *n* = 9; CYP group: *n* = 8. (**B**) Number of cells per high-power field (HPF) for renal immunohistochemistry CD3^+^ and (**C**) CD45R^+^ infiltrate and semiquantitative score for (**D**) F4/80^+^ infiltrate staining. Vehicle group: *n* = 6; Hybri group: *n* = 9; CYP group: *n* = 8. Mean fluorescence intensity for (**E**) glomerular C3 and (**F**) IgG deposition, measured under a confocal microscope. Vehicle group: *n* = 6; Hybri group: *n* = 9; CYP group: *n* = 8. Data are expressed as mean ± S.E.M. ** *p* < 0.01 versus vehicle.

**Figure 3 ijms-23-08411-f003:**
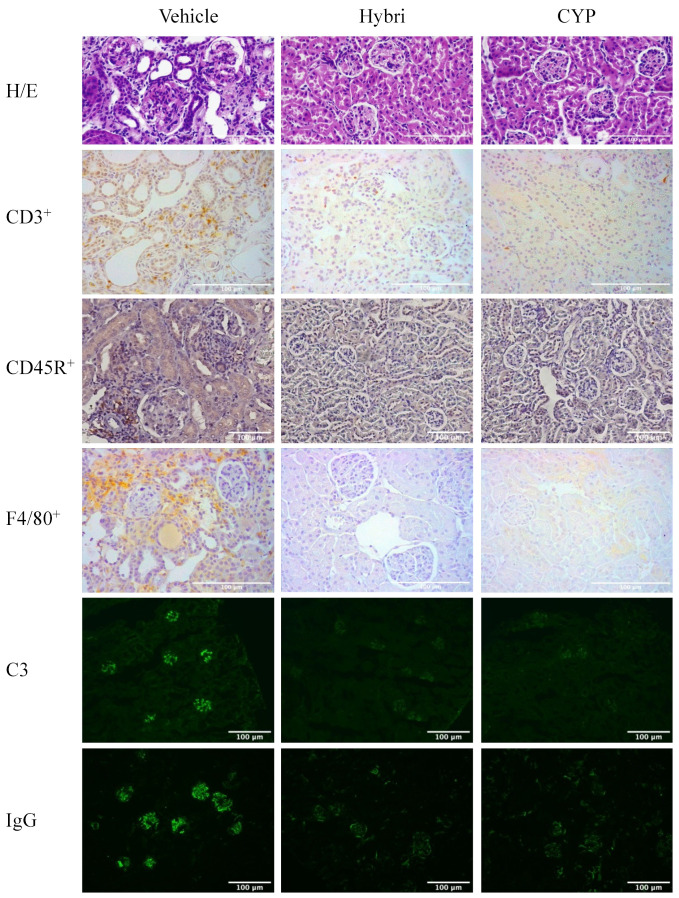
Effect of different treatments on the kidneys in the NZBWF1 study. Representative histological images of hematoxylin and eosin staining and CD3^+^, CD45R^+^, and F4/80^+^ infiltrate using immunohistochemistry. Immunofluorescence images of C3 and IgG deposition.

**Figure 4 ijms-23-08411-f004:**
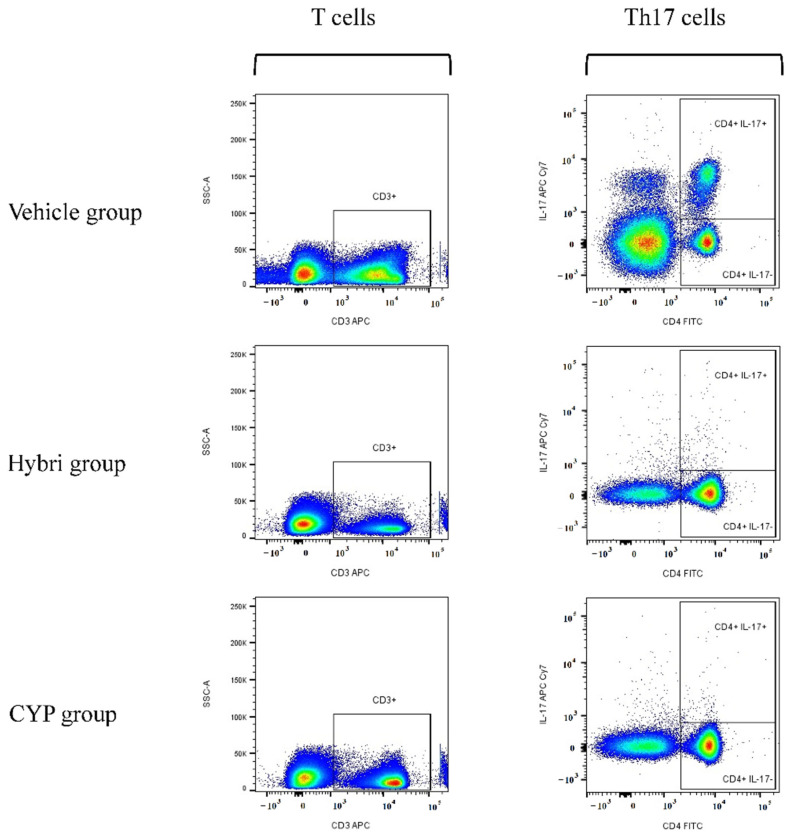
Cumulative graph of Th17 peripheral blood cell analysis at the end of the study, at 36 weeks of age, using flow cytometry software FlowJo ™ v10.8.1. Pseudocolor density plot where blue and green correspond to areas of lower cell density, yellow correspond to mid-range and orange and red are areas of high cell density. CD4^+^ IL-17^+^ population is overexpressed in the vehicle group. Gating strategy for Th17 cells: Obtained events were gated on an FSC intensity and FSC peak dot plot to eliminate doublets. Mononuclear cells were gated on an FSC and SSC dot plot. Lymphocytes were gated on an SSC vs. CD3 dot plot. CD4 and IL-17 dot plot was used to separate CD4 cells with and without IL-17 expression. FSC: Forward scatter, SSC: Side scatter. Down sample of 400,000 events for each group. Vehicle group *n* = 5, Hybri group *n* = 9 and CYP group *n* = 8.

**Figure 5 ijms-23-08411-f005:**
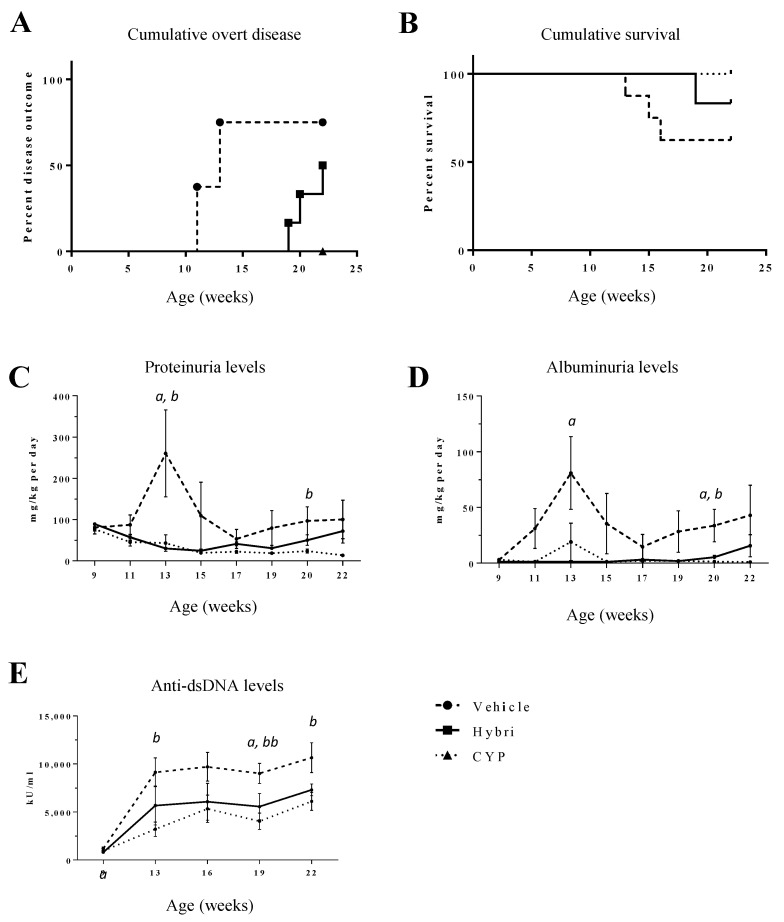
Hybri administration delayed the onset of overt disease, improved the survival curve, and ameliorated functional renal parameters in the MRL/lpr mouse model throughout the study. (**A**) Cumulative overt disease for the MRL/lpr model until the endpoint of the study at 22 weeks of mice age. Disease outcome was considered when proteinuria levels exceeded 3 mg per day. (**B**) Kaplan–Meier survival curve for the MRL/lpr model. Log Rank Mantel–Cox test revealed non-significant differences between groups neither for cumulative overt disease nor cumulative survival. Proteinuria (**C**) and albuminuria (**D**) values throughout the study are expressed in mg normalized per body weight per day. Serum anti-dsDNA (**E**) levels throughout the study are expressed in KU/mL. Vehicle group: *n* = 8; Hybri group: *n* = 6; CYP group: *n* = 6. Data are expressed as mean values ± S.E.M. Hybri vs. Vehicle: *a* = *p* < 0.05, *aa* = *p* < 0.01; CYP vs. Vehicle: *b* = *p* < 0.05, *bb* = *p* < 0.01.

**Figure 6 ijms-23-08411-f006:**
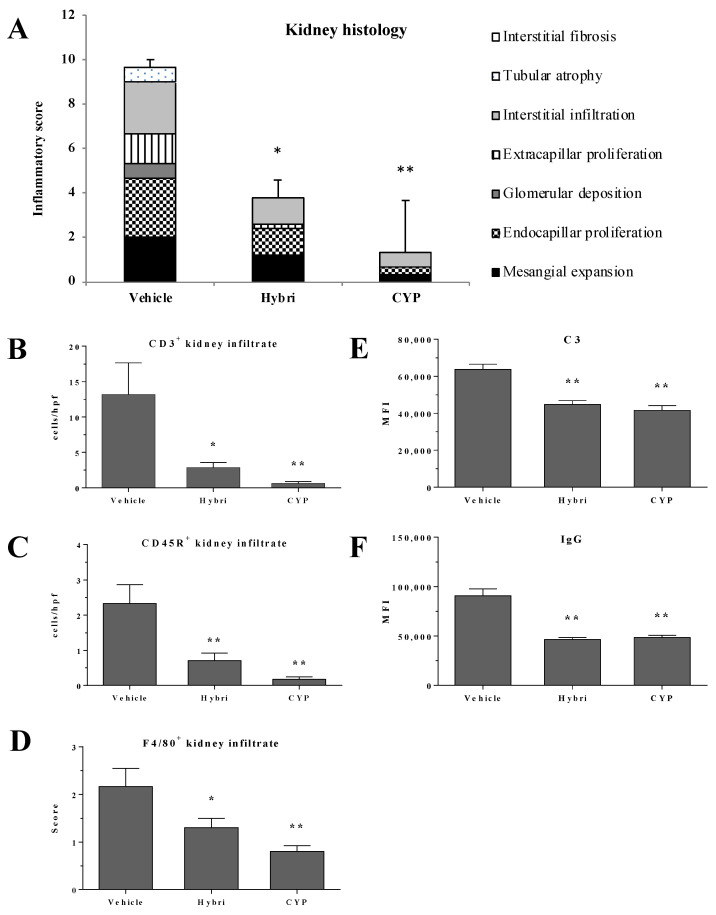
Hybri administration showed an amelioration of kidney morphology and a reduction of effector cell migration and immune deposition in the MRL/lpr mice model. Analysis of samples from surviving mice at the end of the study, at 22 weeks of age. (**A**) Renal histopathology parameters assessed by semiquantitative inflammatory score for hematoxylin and eosin and periodic acid-Schiff staining. Vehicle group: *n* = 3; Hybri group: *n* = 5; CYP group: *n* = 3. (**B**) Number of cells per high-power field (HPF) for renal immunohistochemistry CD3^+^ and (**C**) CD45R^+^ infiltrate and semiquantitative score for (**D**) F4/80^+^ infiltrate staining. Vehicle group: *n* = 5; Hybri group: *n* = 5; CYP group: *n* = 6. Mean fluorescence intensity for (**E**) glomerular C3 and (**F**) IgG deposition, measured under a confocal microscope. Vehicle group: *n* = 5; Hybri group: *n* = 5; CYP group: *n* = 6. Data are expressed as mean ± S.E.M. * *p* < 0.05 versus vehicle; ** *p* < 0.01 versus vehicle.

**Figure 7 ijms-23-08411-f007:**
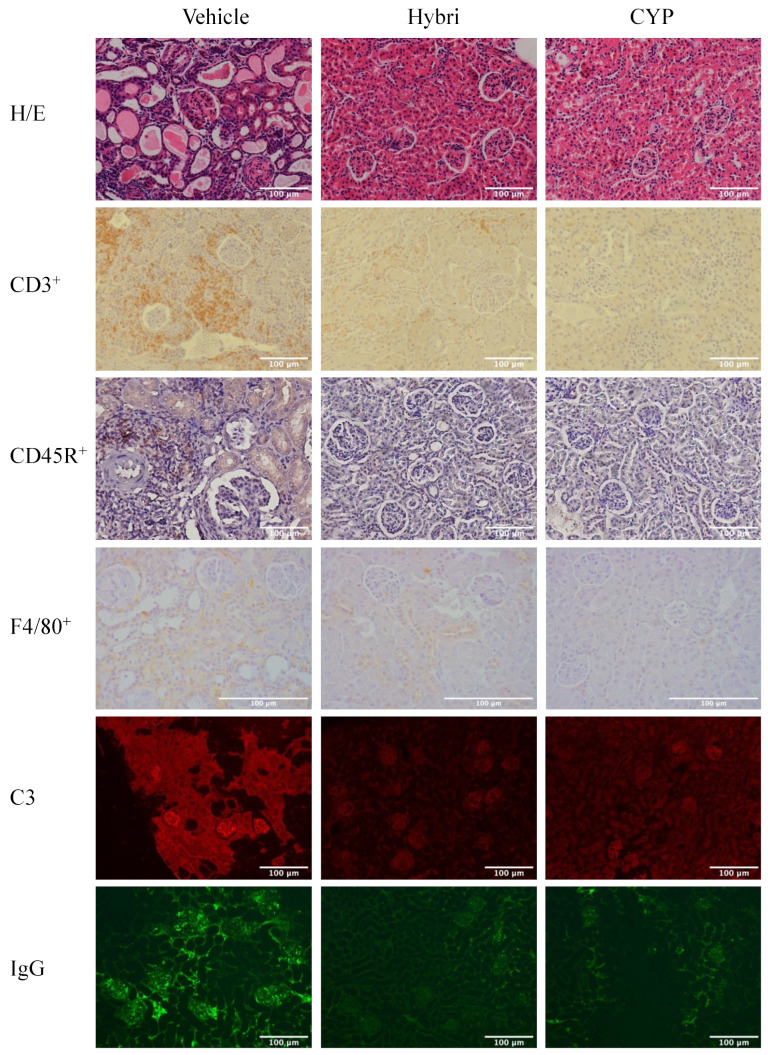
Effect of different treatments on the kidneys in the MRL/lpr study. Representative histological images of hematoxylin and eosin staining and CD3^+^, CD45R^+^, and F4/80^+^ infiltrate using immunohistochemistry. Immunofluorescence images of C3 and IgG deposition.

**Figure 8 ijms-23-08411-f008:**
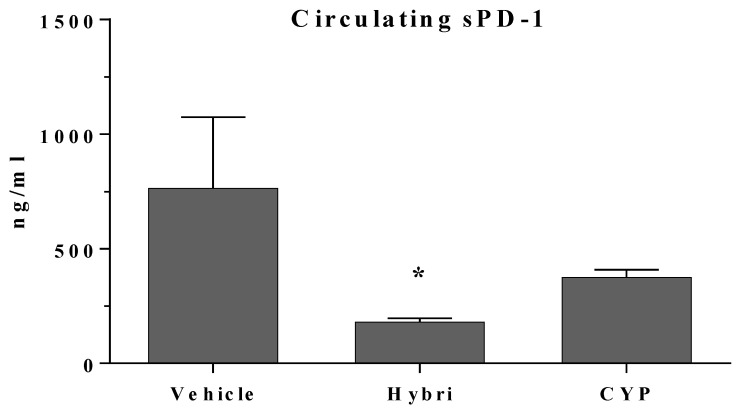
Circulating sPD-1 levels at the end of the study (week 22 of age) were diminished in MRL/lpr mice with Hybri administration. Serum levels of sPD-1 at the end of the second set of experiments in the MRL/lpr mice. Vehicle group: *n* = 5; Hybri group: *n* = 5; CYP group: *n* = 6. Values are expressed in ng/mL and presented as mean ± S.E.M. * *p* < 0.05 versus vehicle.

**Figure 9 ijms-23-08411-f009:**
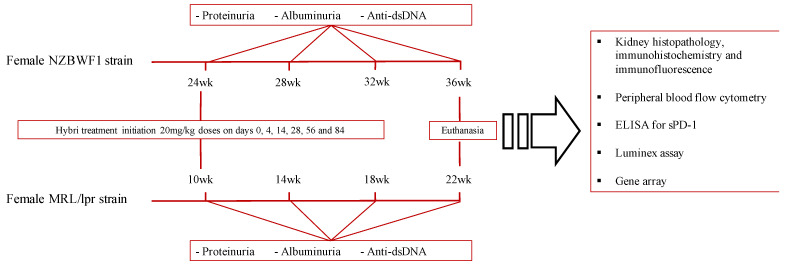
Representative graph of the follow-up of both murine models of lupus nephritis NZBWF1 and MRL/lpr and the analysis techniques performed.

**Table 1 ijms-23-08411-t001:** Peripheral blood cell subsets at the end of the study, at 36 weeks of age in NZBWF1 mouse model analyzed using flow cytometry technique. Results are expressed as percentages ± S.E.M. * *p* < 0.05 versus vehicle; ** *p* < 0.01 versus vehicle.

T-Cell Cytometry Tube	Vehicle	Hybri	CYP
% CD4^+^ in CD3^+^ gating	69.2 ± 1.9	69.2 ± 1.6	64 ± 0.6 *
% PD-1^+^ in CD4^+^ gating	32.1 ± 2.9	39.8 ± 2.9 *	22.3 ± 0.6 *
% CD8^+^ in CD3^+^ gating	24.9 ± 1.9	22.7 ± 1.6	31.4 ± 0.5
% PD-1^+^ in CD8^+^ gating	71.6 ± 2.8	72.4 ± 1.5	70.0 ± 1.1
**Dendritic cell cytometry tube**	**Vehicle**	**Hybri**	**CYP**
% PD-L1^+^ in DC gating	87.8 ± 3.8	87.3 ± 1.6	63.8 ± 1.7 **
% PD-L2^+^ in DC gating	36.3 ± 6.1	51.9 ± 2.9 *	26.2 ± 3.2
% PD-L1^+^ PD-L2^+^ in DC gating	33.8 ± 5.3	49.3 ± 2.8 **	22.5 ± 3.1 *
**Monocyte cytometry tube**	**Vehicle**	**Hybri**	**CYP**
% Stimulated Monocytes in total Monocyte gating	21.8 ± 3.3	29.3 ± 2.4 *	17.5 ± 1.6
% Ly6C^+^ in total Monocyte gating	13.6 ± 2.4	14.9 ± 0.6	20 ± 1.5 **
% Ly6C^+^ in Stimulated Monocyte gating	22.1 ± 4.9	17.2 ± 1	30.4 ± 4.2
**Th17 cytometry tube**	**Vehicle**	**Hybri**	**CYP**
% CD3^+^ CD4^+^ IL-17^+^ cells in CD3^+^ gating	4.2 ± 2.5	0.5 ± 0.1 **	0.2 ± 0.0 **
% CD3^+^ CD4^+^ IL-17^+^ cells in CD4^+^ gating	35.3 ± 21.6	0.6 ± 0.1 **	0.2 ± 0.0 **
**Tregs cytometry tube**	**Vehicle**	**Hybri**	**CYP**
% CD3^+^ CD4^+^ CD25^+^ in CD3^+^ gating	3.1 ± 0.3	3.8 ± 0.2	2.2 ± 0.1 *
% CD3^+^ CD4^+^ CD25^+^ in CD4^+^ gating	8.8 ± 1.2	9.3 ± 0.6	5.5 ± 0.3 **
% Tregs in CD3^+^ gating	2.8 ± 0.3	2.7 ± 0.5	1.9 ± 0.1
% Tregs in CD4^+^ gating	7.9 ± 1.1	6.8 ± 1.5	4.8 ± 0.3
% Tregs in CD25^+^ gating	89.7 ± 1.1	71.0 ± 13.7	88.1 ± 1.9

**Table 2 ijms-23-08411-t002:** Serum cytokine levels at the end of the experiment for all treatments: 36 weeks of age for the NZBWF1 model and 22 weeks of age for the MRL/lpr model. Values are expressed in pg/mL. Data are expressed as mean values ± S.E.M. * *p* < 0.05 versus vehicle; ** *p* < 0.01 versus vehicle.

	NZBWF1 Model			MRL/lpr Model		
	Vehicle	Hybri	CYP	Vehicle	Hybri	CYP
IL-10	67.2 ± 21.6	46.5 ± 7.8	22.6 ± 1.4 *	43.7 ± 3.5	46.2 ± 4.4	22.7 ± 2.7
CD80	5210.1 ± 1848.4	2320 ± 42.3 *	2247 ± 41.1 *	2462.6 ± 1.7	2268 ± 21	2048 ± 59.7
IL-2	50.5 ± 18.9	33.7 ± 4.7	16.3 ± 0.9 *	36 ± 5.1	31.6 ± 5.1	19.6 ± 1.1
IP-10	126.2 ± 45.3	66.6 ± 12.9	54.8 ± 6.6	47.1 ± 13.6	43.3 ± 7.8	39.1 ± 5.6
IL-6	342.6 ± 108.3	174.2± 16.1 *	131.7 ± 6 *	141.2 ± 5.2	143 ± 9.9	141.2 ± 7.9
RANTES	222.9 ± 82.5	92.6 ± 1.4 *	88.3 ± 1.1 *	82 ± 3.9	85.8 ± 4.4	84.4 ± 1.4 *
TNFa	178.4 ± 64.2	114.2 ± 4.6	98.9 ± 1.5	123.7 ± 0.7	105.9 ± 8.2	97.8 ± 1.3
RANKL	623.6 ± 231.1	342.8 ± 19.1	279 ± 13.6 *	205.4 ± 21.7	256.7 ± 14.7	225 ± 13.6
MCP-1	528.1 ± 187.9	213.1 ± 4.2 *	199.3 ± 1.9 *	208.8 ± 2.4	204 ± 1.8	174.1 ± 12.5
IL-17a	87.3 ± 28.2	65 ± 7.2	47.8 ± 3.5	38 ± 4.2	44.7 ± 2.6	44.5 ± 4.6
IL-12	401.3 ± 141.9	157.9 ± 6.5 **	138.7 ± 1.7 **	166.9 ± 12.3	150.7 ± 10.4	131.2 ± 6.4 *
NIG	393.6 ± 88.4	346.5 ± 59.3	189.1 ± 54.9	506.5 ± 40.4	334.9 ± 108.2	206.6 ± 70.7 *
LAG3	897.6 ± 362.9	697.9 ± 197.6	161.6 ± 13.5	5599 ± 2286	1859 ± 460.1 **	192 ± 39.9 **
TIM3	732.3 ± 261.8	580.1 ± 125.2	634.7 ± 121.3	768.6 ± 197.2	344.6 ± 102.9 *	267.8 ± 57.3 **
PDL2	882.1 ± 28.6	1659.5 ± 318.6	577.4 ± 49.6	1795 ± 373.5	596.9 ± 102 **	448.6 ± 39.5 **
TSLP	7.5 ± 1.8	5.2 ± 0.4 *	3.2 ± 0.1 *	6.2 ± 1.6	4 ± 0.3 *	3.8 ± 0.3 *
CD137	21.3 ± 5.4	14.9 ± 1.6	16 ± 1.1	14.5 ± 4.1	7.9 ± 0.3	10.5 ± 1.5

## Data Availability

The data presented in this study are available on request from the corresponding author.

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
