# Peer review of "Dual Costimulatory and Coinhibitory Targeting with a Hybrid Fusion Protein as an Immunomodulatory Therapy in Lupus Nephritis Mice Models"

_ijms, 2022, doi:10.3390/ijms23158411_

Round 1

Reviewer 1 Report

The research team worked hard, but unfortunately the design of the study was not successful.

It seems as if they didn't do it just now, a couple of years of work, with an almost updated references.

My comments: 

1.Based on literature data was the study design appropriate? Why did you use Hibry alone?

Based on the results obtained, what makes you think that Hibry can be an alternative treatment in lupus nephritis? Not better than cyclophosphamide.

2. The discussion needs to be reconsidered.

3. The introduction is too long. The 65-82 line is almost redundant. 

4. References are very old. For example the reference to pathogenesis and therapy (1-4). All of the reference should be updated.

5. the figure 3A and 5A: it would be better if it were in colorful or patterned. The figures, especially number 1, number 2 and number 5, are not very expressive, these should be reconsidered.

6. It would be nice to have an illustration of the mechanism of action of Hybrid

Author Response

The research team worked hard, but unfortunately the design of the study was not successful.

It seems as if they didn't do it just now, a couple of years of work, with an almost updated references.

My comments:

1.Based on literature data was the study design appropriate? Why did you use Hibry alone?

We appreciate the reviewer’s comments as they are very accurate.

We think that the design of the study is appropriate because, beyond using a model widely used in these types of studies, the administration of Hybri alone is mandatory to observe its therapeutic potency and protective effect.

We wanted to inoculate only Hybri so that we could attribute the therapeutic benefits only to the protein. In the case of concomitant treatment, it would be difficult and perhaps too theoretical to demonstrate what effects can be attributed to Hybri.

In a later direction we can try to modify the doses and vary the administration regimen, as well as administer Hybri concomitantly with another agent.

Based on the results obtained, what makes you think that Hibry can be an alternative treatment in lupus nephritis? Not better than cyclophosphamide.

The idea of using Hybri in lupus nephritis was precisely due to the long-term toxicity observed in the clinic using cyclophosphamide. Despite knowing that this exists in humans, in our short-lived pre-clinical murine model this effect was not observed. We assume that this toxicity is important to consider developing other agents with fewer adverse effects.

Although we have not yet done toxicity studies for Hybri, we have planned to perform in a near future in the road to introduce the compound in the clinics.

We don’t expect that our compound may induce toxicity itself other beyond than the derived from its mechanism of action, as it is a completely human recombinant protein. Biological agents usually have better predictions in toxicity studies rather than small molecules or xenobiotics. Otherwise, animals from our study receiving Hybri didn’t show any signal of side effects.

Finally, although not better than cyclophosphamide, the observed therapeutic effects were similar in most parameters and altogether significantly better than in the Vehicle group.

  1. The discussion needs to be reconsidered.

We have modified some concepts in the discussion, given the requirements of the reviewers.

  1. The introduction is too long. The 65-82 line is almost redundant.

We have reduced the paragraph you mentioned. We agree that this part of the introduction does not announce any relevant information.

  1. References are very old. For example the reference to pathogenesis and therapy (1-4). All of the reference should be updated.

We have updated some references of the manuscript.

  1. the figure 3A and 5A: it would be better if it were in colorful or patterned. The figures, especially number 1, number 2 and number 5, are not very expressive, these should be reconsidered.

Regarding Figures 3A and 5A, we used another pattern for better visual comprehension.

By Figures 1 2 and 5, we have reconsidered and we believe that they should be maintained, as the study is based on the observation of pathological improvements not only in mortality but in renal function and autoantibody production.

In any case, we redistributed the graphs to finally get only one figure for each study, as a proposal (now Figures 1 and 5).

  1. It would be nice to have an illustration of the mechanism of action of Hybrid

We have uploaded a figure of the postulated potential mechanism of action of Hybri on the resubmission. Our new proposal is to add it as a Graphical Abstract. The ancient Graphical Abstract is transformed to a figure of Methods section (Figure 9).

Reviewer 2 Report

The manuscript titled “Dual costimulatory and coinhibitory targeting with a hybrid fusion protein as an effective immunomodulatory therapy in lupus nephritis” described a rigorous and well described sets of studies aimed to preventing or delaying onset of nephritis in two well-known and established mouse models of lupus nephritis. Investigators found a significant benefit approaching that of IP cyclophosphamide. Investigators should be acknowledged for use of positive and negative controls for their experiments.

Major concerns that decrease potential impact of the manuscript:

[1] The title could be misunderstood. Please change to remove any suggestion that the experimental therapy is “effective” as the study is not a clinical trial but a pre-clinical study. Also, please state in the title that you are using a lupus nephritis model to clearly indicate that this study is not human subjects research.

[2] It is unfortunate that the protocol dictated stopping the experiment with only 25% mortality in the vehicle control group, and that their specific strain of NZBWF1 mice seem to have less severe disease by 9 months (compared to historical control mice that have 50% mortality by this age). Furthermore, following survival of treated mice for another 3 months to 12 months of age would have helped to determine whether survival is just prolonged or whether mice are truly protected from early mortality. Similarly, following the treated MRL/lpr mice another month might have better shown the extent of delayed mortality in the treated mice.

[3] Please clearly state in methods at what age mice were euthanized for kidney histopathology assessment. Similarly, methods and/or figure legends should specifically state the age of the mice at time of analysis (peripheral blood cell immunophenotyping, Luminex, and TaqMan qRT-PCR). The discussion should also include a statement that histopathology/changes in transcriptome in the vehicle control group could be underestimated, due to the fact that those mice with the worst nephritis likely were those who died prior to euthanasia and histopathologic assessment. If histopathology for mice that died early were included in the analyses, this should be specifically stated in the methods.

[4] Supplemental figure 2: the immunohistochemical staining for CD3 and F4/80 is very faint. Can images be brightened for better illustration of signal to noise ratio, and to allow assessment of glomerular vs tubulointerstitial staining patterns. Also, the IF staining for IgG could be brighter for all 3 groups to better compare degree of staining. Otherwise, the images are representative of the quantitative data and allow the reader to assess the distribution of staining. This reviewer might suggest including this figure in the manuscript and not keeping it supplemental.

[5] Why is soluble PD1 expression only assessed in MRL model? This begs the question of whether the experiment was performed in the NZBWF1 mice and not included in the manuscript for some reason.

Minor issues / suggestions:

[6] Introduction, page 1, line 45: it is no longer true that the only available therapies for lupus are non-specific. Belimumab is an approved agent targeting B-LyS, anifrolumab is an approved agent targeting the receptor for alpha interferon. Arguably, voclosporin could be considered a targeted therapy for T-cells.

[7] Introduction, page 3, line 99-100: Please expand your claim that targeting PD1 can prevent SLE. The titles of the cited references do not clearly indicate that there is evidence to back up this statement. Is there evidence beyond just correlation between disease activity and soluble PD1 levels specifically in SLE or lupus models?

[8] Introduction, page 3, line 114: What are the binding affinities of the novel fusion protein to CD80 and PD1? Are the binding affinities the same when both targets are bound instead of each individually? How do these binding affinities compare to affinities of natural CTLA4, CD28, and PD-L2? Please include in introduction so that readers do not need to look up this information in your prior manuscript.

[9] Methods should state why the dose of 20mg/kg was selected. The rationale for dosing interval is clearly stated. Discussion should emphasize that future directions will need to include dose response curves for the fusion protein, and or more frequent dosing, to determine whether nephritis responses can be as effective as IP cyclophosphamide (as written, it is buried and might be missed by a casual reader).

[10] Methods should include a rationale for the numbers of mice in each group. Was a power analysis done a priori? 

[11] Figure 3: It is nice proof of principle to see that mesangial expansion and endocapillary hypercellularity is not affected by treatment of mice with the fusion protein. This would not be expected with a targeted therapy specific for lymphocytes. It would be nice in the discussion to include some of the data on costimulatory molecule expression on renal parenchymal cells. At least podocytes have been shown to express CD80 and future directions could include in vivo and in vitro studies looking at whether the fusion protein binds to glomerular cells.

[12] Table 1: it is unclear why there would be increased PD-L2 expression on APC in the mice treated with fusion protein. Can authors speculate in discussion?

[13] Discussion, page 15, line 481-3: The microarray data on kidney tissue does not actually support the statement that “modify inflammatory and immunomodulatory pathways in the renal parenchyma.” Typically, parenchymal cells are distinguished from inflammatory cells as those in the kidney that derive from mesenchymal tissues rather than from bone marrow. The sentence should be revised to state that “modify inflammatory and immunomodulatory pathways in the kidney.”  Most of the genes assessed could be differentially expressed secondary to changes in degrees of inflammation.

Author Response

The manuscript titled “Dual costimulatory and coinhibitory targeting with a hybrid fusion protein as an effective immunomodulatory therapy in lupus nephritis” described a rigorous and well described sets of studies aimed to preventing or delaying onset of nephritis in two well-known and established mouse models of lupus nephritis. Investigators found a significant benefit approaching that of IP cyclophosphamide. Investigators should be acknowledged for use of positive and negative controls for their experiments.

 We appreciate the reviewer’s initial feedback and comments as they are very accurate.

Major concerns that decrease potential impact of the manuscript:

[1] The title could be misunderstood. Please change to remove any suggestion that the experimental therapy is “effective” as the study is not a clinical trial but a pre-clinical study. Also, please state in the title that you are using a lupus nephritis model to clearly indicate that this study is not human subjects research.

We completely agree with this observation and we have modified the title of the manuscript according to your suggestion:

“Dual costimulatory and coinhibitory targeting with a hybrid fusion protein as an immunomodulatory therapy in lupus nephritis mice models”

[2] It is unfortunate that the protocol dictated stopping the experiment with only 25% mortality in the vehicle control group, and that their specific strain of NZBWF1 mice seem to have less severe disease by 9 months (compared to historical control mice that have 50% mortality by this age). Furthermore, following survival of treated mice for another 3 months to 12 months of age would have helped to determine whether survival is just prolonged or whether mice are truly protected from early mortality. Similarly, following the treated MRL/lpr mice another month might have better shown the extent of delayed mortality in the treated mice.

We understand that it may be considered unfortunate to obtain low mortality at the end of the study, as studies with higher mortality in the untreated group can be found in literature. This NZBWF1 strain is widely used in animal studies of lupus nephritis. However, this strain presents a certain heterogeneity in the development of the disease and makes it difficult to predict disease outcome and mortality when defining a protocol, a tracking of mice or the amount of protein needed to synthesize.

Present schedule is widely used everywhere and our group has used it routinely, giving us several publications. It is worth noting that some authors end the follow-up in week 32 of mice life, but in our study, we preferred to extend the follow-up until week 36 precisely to be able to observe a more advanced state of the disease in the Vehicle group. However, we aimed to perform a concept study rather than a mortality analysis. We wanted to get enough biological samples from Vehicle group so we didn’t look for a higher mortality. There are other more essential parameters to evaluate the Hybri therapeutical effect.

However, our group is considering to perform a rescue study starting treatment later, when the disease is more advanced.

[3] Please clearly state in methods at what age mice were euthanized for kidney histopathology assessment. Similarly, methods and/or figure legends should specifically state the age of the mice at time of analysis (peripheral blood cell immunophenotyping, Luminex, and TaqMan qRT-PCR). The discussion should also include a statement that histopathology/changes in transcriptome in the vehicle control group could be underestimated, due to the fact that those mice with the worst nephritis likely were those who died prior to euthanasia and histopathologic assessment. If histopathology for mice that died early were included in the analyses, this should be specifically stated in the methods.

We have modified the methods section according to your suggestions (lines 571-573 of the track changes document). We have added the age at which the mice were euthanized, both in the methods section and in the histopathology, peripheral blood flow cytometry, Luminex, serum sPD-1 and gene array figures. We have also added in the methods a figure of the study design for a better understanding (Figure 9, previously the Graphical Abstract).

We also appreciate the concept you propose referring to dead animals before the end of the study. We added this sentence in the discussion, as you proposed (lines 541-544 of the track changes document).

In this study no samples of dead animals were used in any technique performed, not even in histopathology.

[4] Supplemental figure 2: the immunohistochemical staining for CD3 and F4/80 is very faint. Can images be brightened for better illustration of signal to noise ratio, and to allow assessment of glomerular vs tubulointerstitial staining patterns. Also, the IF staining for IgG could be brighter for all 3 groups to better compare degree of staining. Otherwise, the images are representative of the quantitative data and allow the reader to assess the distribution of staining. This reviewer might suggest including this figure in the manuscript and not keeping it supplemental.

We modified the Figure S2 as you proposed. We have applied the same modifications for each group of terms of brightness to improve the contrast of the CD3 and F4/80 images for better illustration. Now the compartments and staining look better. The IgG IF has also been improved to better compare the degree of staining.

Thank you for the suggestion. We have moved Figures S2 and S3 into the manuscript (now Figures 3 and 7).

[5] Why is soluble PD1 expression only assessed in MRL model? This begs the question of whether the experiment was performed in the NZBWF1 mice and not included in the manuscript for some reason.

As we say in the manuscript, the good response promoted by Hybri in the first NZBWF1 model led us to test the protein as a treatment in a more complex model such as MRL/lpr. To choose this second model, we observed in the literature that it is more virulent and has other effects depending on other pathways, such as rheumatoid factor.

Looking further into the literature we observed the correlation between soluble PD-1 and the progression of rheumatoid arthritis, as well as its relationship to this strain of mice. For this reason and at that time, we wanted to add this measurement to this new model.

It is true that this analysis makes some sense in the NZBWF1 study, as PD-1 is probably linked to Hybri's PD-L2, but beyond that we did not find enough evidence in the reviewed literature to assume that it is a mechanism related to the pathogenesis of the disease in this particular strain. Also, we hadn’t enough serum samples at that point.

Minor issues / suggestions:

[6] Introduction, page 1, line 45: it is no longer true that the only available therapies for lupus are non-specific. Belimumab is an approved agent targeting B-LyS, anifrolumab is an approved agent targeting the receptor for alpha interferon. Arguably, voclosporin could be considered a targeted therapy for T-cells.

You are absolutely right. We have modified the introduction based on what you have commented. We have removed the sentence from lines 45 to 48 and redone the paragraph from line 94 to 106 of the track changes document.

[7] Introduction, page 3, line 99-100: Please expand your claim that targeting PD1 can prevent SLE. The titles of the cited references do not clearly indicate that there is evidence to back up this statement. Is there evidence beyond just correlation between disease activity and soluble PD1 levels specifically in SLE or lupus models?

As you rightly say, we have not found literature that directly identifies as crucial the PD-1 pathway in SLE beyond the correlation between sPD-1 levels and disease activity. Rheumatoid arthritis is perhaps more studied, but in the case of human SLE, there is only a correlation that can lead to a theoretical assumption.

We have modified the sentence by changing the concept: “The PD-1 signaling pathway has recently been suggested as a crucial target to prevent the development of SLE.”

[8] Introduction, page 3, line 114: What are the binding affinities of the novel fusion protein to CD80 and PD1? Are the binding affinities the same when both targets are bound instead of each individually? How do these binding affinities compare to affinities of natural CTLA4, CD28, and PD-L2? Please include in introduction so that readers do not need to look up this information in your prior manuscript.

Hybri’s affinities with its murine ligands (CD80 and PD-1) can be found in our group’s previously published article. In this article we showed their affinities and that neither domain affects the binding affinity of the other. We have added this information in the manuscript introduction (line 131 to 133 of the track changes document). There is no further information to add from the previous article regarding the affinity of protein domains.

As for the comparison with CTLA4, CD28 and PD-L2, so far, we have only compared internally with the existing literature. In some cases, we have found literature showing binding affinities between human proteins, which do not allowes to extrapolate with binding affinities between human and murine proteins due to the approximately 70% of protein homology between species, as is the case in our previous published work.

We are currently scheduled for an upcoming study with human CD80 and PD-1 to assess Hybri affinities. Also, we planned to asses human CTLA4, PD-L2, and CD28 to human CD80 and PD-1 in order to compare with Hybri affinities.

[9] Methods should state why the dose of 20mg/kg was selected. The rationale for dosing interval is clearly stated. Discussion should emphasize that future directions will need to include dose response curves for the fusion protein, and or more frequent dosing, to determine whether nephritis responses can be as effective as IP cyclophosphamide (as written, it is buried and might be missed by a casual reader).

We appreciate the suggestion.

The rationale for using 20mg/kg was based on the empirical assumption of administering a similar molar dosage of CTLA4 as used in the clinics. That is, Abatacept dosage used is 10 mg/kg. Since the molecular weight of Hybri is about twice that of Abatacept, we used twice the weight of Hybri to get the same amount of Abatacept.

Moreover, in our previous study of renal ischemia-reperfusion injury, we used this dose, which was effective. We have added the information in the Methods section (lines 557-559 of the track changes document).

As you say, future directions are dose response curves and possibly testing more doses or varying dose frequencies. We have made this part more explicit in the discussion (lines 436-438 of the track changes document).

[10] Methods should include a rationale for the numbers of mice in each group. Was a power analysis done a priori?

We did not do an a priori statistical power analysis.

We are conscious that the number of animals used for the studies is relatively low, especially if we compare with clinical trials where the researcher has to calculate the necessary “n” of patients. However, animal ethics committees vehemently insist on the Principles of Humane Experimental Techniques, which are based on avoiding the excessive suffering of animals if this is not strictly necessary. It refers to looking for alternatives for reducing, replacing and refining experimental animal models in order to limit the use of animals.

[11] Figure 3: It is nice proof of principle to see that mesangial expansion and endocapillary hypercellularity is not affected by treatment of mice with the fusion protein. This would not be expected with a targeted therapy specific for lymphocytes. It would be nice in the discussion to include some of the data on costimulatory molecule expression on renal parenchymal cells. At least podocytes have been shown to express CD80 and future directions could include in vivo and in vitro studies looking at whether the fusion protein binds to glomerular cells.

Thank you very much for your beautiful comments. We agree with your suggestions.

We have added this sentence and some data on costimulatory molecule expression in renal parenchyma, as you can see in the discussion (lines 447-453 of the track changes document).

[12] Table 1: it is unclear why there would be increased PD-L2 expression on APC in the mice treated with fusion protein. Can authors speculate in discussion?

We have added a comment to the discussion (lines 466-471 of the track changes document) speculating in the direction we expect is the potential effect of Hybri in this setting.

[13] Discussion, page 15, line 481-3: The microarray data on kidney tissue does not actually support the statement that “modify inflammatory and immunomodulatory pathways in the renal parenchyma.” Typically, parenchymal cells are distinguished from inflammatory cells as those in the kidney that derive from mesenchymal tissues rather than from bone marrow. The sentence should be revised to state that “modify inflammatory and immunomodulatory pathways in the kidney.”  Most of the genes assessed could be differentially expressed secondary to changes in degrees of inflammation.

You are absolutely right with this comment. We have modified this sentence in accordance with this rational (lines 528-529 of the track changes document). Thank you.